# Impact of Exercise Interventions on Sustained Brain Health Outcomes in Frail Older Individuals: A Comprehensive Review of Systematic Reviews

**DOI:** 10.3390/healthcare11243160

**Published:** 2023-12-13

**Authors:** Guilherme Eustáquio Furtado, Anne Sulivan Lopes da Silva Reis, Ricardo Braga-Pereira, Adriana Caldo-Silva, Pedro Teques, António Rodrigues Sampaio, Carlos André Freitas dos Santos, André Luís Lacerda Bachi, Francisco Campos, Grasiely Faccin Borges, Sónia Brito-Costa

**Affiliations:** 1Polytechnic Institute of Coimbra, Applied Research Institute, Rua da Misericórdia, Lagar dos Cortiços-S. Martinho do Bispo, 3045-093 Coimbra, Portugal; sonya.b.costa@gmail.com; 2Research Centre for Natural Resources Environment and Society (CERNAS), Polytechnic Institute of Coimbra, Bencanta, 3045-601 Coimbra, Portugal; 3Postgraduate Program in Physical Education, University of Southwest Bahia and the State University of Santa Cruz (PPGEF/UESB/UESC), Ilhéus 45650-000, Brazil; diploanne@gmail.com; 4N2i, Research Centre of the Polytechnic Institute of Maia, 4475-690 Maia, Portugal; rp88st@gmail.com (R.B.-P.); pteques@ipmaia.pt (P.T.); arsampaio@umaia.pt (A.R.S.); 5Research Centre for Sport and Physical Activity, CIDAF, Faculty of Sport Science and Physical Education, 3040-248 Coimbra, Portugal; dricaldo@gmail.com; 6CIPER, Interdisciplinary Center for the Study of Human Performance, 1499-002 Lisbon, Portugal; 7Discipline of Geriatrics and Gerontology, Department of Medicine, Paulista School of Medicine, Federal University of Sao Paulo (UNIFESP), São Paulo 04020-050, Brazil; carlos.freitas@unifesp.br; 8Postgraduate Program in Translational Medicine, Department of Medicine, Paulista School of Medicine, Federal University of São Paulo (UNIFESP), São Paulo 04023-062, Brazil; 9Post-Graduation Program in Health Sciences, Santo Amaro University (UNISA), São Paulo 04829-300, Brazil; 10Coimbra Education School, Polytechnic of Coimbra, 3045-043 Coimbra, Portugal; francicampos@esec.pt; 11Center for Public Policies and Social Technologies, Federal University of Southern Bahia, Praça José Bastos, s/n, Centro, Itabuna 45600-923, Brazil; grasiely.borges@cpf.ufsb.edu.br; 12Research Group in Social and Human Sciences (NICSH), Coimbra Education School, Polytechnic of Coimbra, 3045-043 Coimbra, Portugal; 13Human Potential Development Center (CDPH), Polytechnic of Coimbra, 3030-329 Coimbra, Portugal

**Keywords:** quality of evidence, pre-frail, dementia, emotional well-being, multicomponent exercise, muscle strength, sustainability development goals

## Abstract

Several systematic review studies highlight exercise’s positive impact on brain health outcomes for frail individuals. This study adopts a Comprehensive Review of reviews (CRs) approach to amalgamate data from existing reviews, focusing on exercise’s influence on brain health outcomes in older frail and pre-frail adults. The methodology involves a thorough search of Portuguese, Spanish, and English-indexed databases (i.e., Ebsco Health, Scielo, ERIC, LILACS, Medline, Web of Science, SportDiscus) from 1990 to 2022, with the AMSTAR-2 tool assessing evidence robustness. The search terms “physical exercise”, “elderly frail”, and “systematic review” were employed. Results: Out of 12 systematically reviewed studies, four presented high-quality (with metanalyses), while eight exhibit critically low quality. Positive trends emerge in specific cognitive and neuromotor aspects, yet challenges persist in psychosocial domains, complex cognitive tasks, and ADL outcomes. This study yields reasonable and promising evidence regarding exercise’s influence on quality of life and depression in frail older individuals. However, the impact on biochemical markers remains inconclusive, emphasizing the need for standardized methodologies. Conclusions: The findings highlight the importance of acknowledging methodological nuances for clinicians and policymakers when translating these results into impactful interventions for aging populations. This emphasizes the necessity for a comprehensive and customized approach to exercise interventions aimed at fostering the sustainability of overall well-being in older individuals, aligning with United Nations Sustainable Development Goal 3.

## 1. Introduction

In the ever-evolving landscape of aging and health, the complex relationship between physical well-being and cognitive vitality has become a focal point for researchers and practitioners [1]. Within the context of advancing age, the preservation of brain health takes on heightened significance, particularly in the domain of frailty [2]. This exploration delves into the hypothesis of exercise-induced health-related positive changes, aiming to elucidate their profound impact on sustaining brain health domains among older populations.

Contemporary perspectives on frailty view it as a holistic syndrome intertwined with environmental, psychosocial, and behavioral factors, collectively contributing to premature frailty onset [3,4]. The assessment of frailty syndrome (FS) has evolved to encompass a spectrum of tools, ranging from one-dimensional to multidimensional measures [5], including physical [6,7], psychometric [8,9], and multi-methods [10]. The emergence of “overlapping syndromes” and correlations associated with FS has brought attention to subgroups [11,12,13,14,15,16], such as cognitive frailty, marked by concurrent physical frailty and potentially reversible cognitive impairment [17,18].

The intersection of brain health and FS has garnered considerable attention, revealing the intricate interplay among cognitive, sensory, social-emotional, behavioral, and neuromotor domains [19,20]. As per the World Health Organization (WHO), brain health is defined by indicators reflecting the dynamic status of various domains within older individuals [19]. This connection has unveiled links between frailty and brain health, emphasizing the reciprocal influence between these vital aspects of aging [21,22,23,24,25].

Given the varied approaches to frailty syndrome (FS) assessment, there is a consensus that a sedentary lifestyle significantly contributes to its development [26]. In recent years, exercise interventions have been shown to reduce stress, anxiety, and the risk of neurobiological and psychological disorders, all closely associated with FS [27,28,29,30]. Exercise-based interventions offer promise for addressing FS [31], presenting opportunities for biopsychological modulation with positive effects on several factors linked to FS [32]. Consequently, research on frail older individuals has expanded its focus to encompass neurocognitive, psychosocial, and behavioral dimensions alongside specific FS variables [33].

While several systematic reviews (SRs) have examined the impact of exercise interventions on brain health-related frailty outcomes [34], a comprehensive review of SRs (CRs) becomes imperative to synthesize this wealth of information [35]. A CRs offers a logical progression, providing a comparative analysis of individual reviews to furnish decision-makers with a unified evidence base [36]. Recent examinations underscore the invaluable role of CRs in several aspects, including the effects of exercise on cancer, acute low back pain, depression, dementia and hemophilia [37,38,39,40,41,42]. Notably, a scarcity of CRs is evident in the context of exercise interventions for frailty [43].

This CRs was designed to provide a panoramic understanding of the evidence landscape in the specific paradigm of exercise-induced positive changes in brain health outcomes. Unlike conventional methodologies [44], our inclusive approach aims to offer valuable insights into overall trends, gaps, and consistencies across multiple SRs. This strategy aligns with the study’s goal of synthesizing a comprehensive body of evidence, contributing to a more robust understanding of the research landscape. It seeks to identify primary types of exercise interventions, evaluate their efficacy, and highlight extensively researched variables related to brain health in this context. Recognizing that evidence may vary for different indicators of brain health, it aims for a more objective understanding of these aspects.

In an exploratory manner, this article also delves into elements related to the sustainability of exercise interventions on brain health outcomes, aligning with the United Nations Sustainable Development Goals (SDGs). The relationships between physical exercise, brain health, and sustainability are examined based on variables inherent to the conduct of physical exercise programs, such as study design, program duration, intensity, frequency, and stability of positive results. This exploratory dimension adds a critical perspective to the broader discourse on the long-term impact and viability of exercise interventions in promoting brain health among older individuals.

## 2. Method

The methodology employed in conducting these CRs is rooted in the approach outlined by Smith and colleagues [36]. This method is designed to assess and amalgamate research outcomes from numerous SRs papers into a cohesive document. Its objective is to systematically compare and contrast diverse findings, providing clinicians and policy decision-makers with relevant scientific evidence. The primary areas of focus include literature sourcing and searching, review selection, quality appraisal, result presentation, and outlining practical research implications.

### 2.1. Search Strategy

This study adhered to a predetermined and publicly available protocol documented in PROSPERO 2022 under registration number CRD42018088782 [45]. The search process was conducted in English across an array of databases: Ebsco, Ebsco Health, Scielo, ERIC, LILACS, Medline (PubMed), Web of Science, SportDiscus, and the Joanna Briggs Institute Database. The online remote-control system of the University of Coimbra was accessed [46]. The data collection occurred between December 2021 and February 2022, using the advanced meta-search feature to identify relevant reviews and meta-analyses published between 1990 (the year of the first works in the frailty field) and 2022. To enhance the precision of the search, Medical Subject Headings (MeSH) indexed descriptors [47], as presented in Table 1, along with corresponding terms, were identified and incorporated into the search strategy.

### 2.2. Inclusion Criteria

The search process adhered to the PICOTS (Patient, Intervention, Comparison, Outcome, Time, Study Design/Settings) framework to guide its trajectory [48,49]. Table 2 presents an overview of the key concepts used to guide the meta-search according to the PICOTS guidelines. Our research group has employed this practice to enhance the search accuracy and the reader’s understanding of the core topics of the SRs. In general, this study encompassed SRs that included participants of both genders, aged 60 years and above, classified as pre-frail or frail, residing in community settings or 24 h healthcare and social support centers. Categorization was executed employing robust and statistically validated frailty protocols [5,50,51].

The “intervention” concept encompassed several methods of supervised group-based physical exercise interventions. This encompassed classic modalities like resistance training, muscle strengthening, and flexibility exercises, alongside contemporary approaches like yoga, Pilates, and multicomponent and/or chair-based exercise programs. Relevant details regarding the average intervention duration and frequency were documented.

The analysis incorporated “comparison or control conditions”, encompassing systematic reviews (SRs) that included both randomized (RCTs) and non-randomized trials, with and without control groups. The assessment of outcomes spanned a comprehensive range of brain health domains, aligning with the conceptual framework proposed by the WHO. [19]. These included psychosocial aspects, psychological well-being, quality of life, cognitive capacities, sleep behaviour, and neurological as well as neuromotor functions.

“Timing” pertains to the nuanced consideration of the duration, frequency, and temporal aspects of interventions or exposures under study. It involves precisely specifying the timeframes during which interventions or exposures are administered and assessing their potential short-term and long-term effects. “Settings” encompass the diverse environments or contexts in which interventions, exposures, or measurements occur. This includes factors such as geographical locations, healthcare facilities, community settings, or virtual platforms where the study is conducted [48].

### 2.3. Exclusion Criteria

The following criteria were applied for excluding SR articles from this CR: (i) studies that primarily focused on older patients residing in acute care and hospital settings; (ii) articles that did not present any assessment of brain health outcomes non-adherence to established validity criteria to the operationalized definition for frailty screening; (i.e., psychological, neuromotor, cognitive); (iii) non-specific target on frail populations; (iv) exercise interventions that were not supervised and lacked a group-based approach; (v) therapeutic interventions or other similar group therapies, not characterized as supervised physical exercise programs with a group-based approach; (vi) multifactorial interventions such as nutritional, pharmacological agents, and others; (vii) non-adherence to established validity criteria or utilization of an operationalized definition for frailty screening.

### 2.4. Screening of the Quality of Evidence

The robustness of the evidence presented in each article was evaluated using the AMSTAR-2 (A Measurement Tool to Assess Systematic Reviews) [52]. The AMSTAR-2 is a validated instrument designed to assess the quality of the best available evidence within SRs [36]. Through an online calculator, this tool evaluates 16 distinct elements and generates a comprehensive outcome based on both quantitative scoring (ranging from 0 to 16 points) and qualitative classification (ranging from critical, low, moderate, to high quality). Given its relevance in delving into the methodology and quality of systematic reviews, AMSTAR-2 serves as a means to mitigate potential errors and biases [37]. Each paper considered during the eligibility phase underwent assessment using this tool (for additional details, refer to: https://amstar.ca/Amstar_Checklist.php, (accessed on 4 May 2023). Articles were excluded from consideration for critical quality of evidence only when essential details were lacking, such as clarity in reporting, conflicts of interest, insufficient details in this study design section, and relevance to the research question. This exclusion was in accordance with the previously outlined method for selecting systematic review articles [52].

### 2.5. Data Extraction and Synthesis

The search process and final article selection adhered to the guidelines outlined in the Preferred Reporting Items for Systematic Reviews and Meta-Analyses (PRISMA) statement [53]. Initial searches were conducted by two distinct researchers, utilizing a predefined set of keywords. The relevance of articles from the original search was determined through the following screening stages:(a)Identification Phase: Titles and abstracts were reviewed, and relevant citations were saved using specialized software. Duplicate entries were removed during the second screening phase.(b)Screening Phase: Full-text reading was performed for articles that passed the identification phase. If an article did not align with the inclusion criteria during the advanced eligibility phase, it was excluded from the study.(c)Included Phase: Manuscripts that underwent AMSTAR assessment and demonstrated critical quality evidence were excluded if essential information was missing [52].(d)Data concerning the impact of interventions on mental health outcomes were extracted from selected systematic reviews and organized in tabular form. This data were further supplemented by a narrative synthesis aimed at addressing the review question.

Data extraction from the included SR papers in this CSR involved gathering information related to the PICOTS statement and associated details: (i) Basic characteristics of the included SRs; (ii) Number of studies (k) and participants (n) for each study; (iii) Research design and intervention types; (iv) Brain health variable categories and the observed impact of interventions.

### 2.6. Elements Related to the Sustainability of Exercise Intervention Programs

The assessment of elements related to the sustainability of exercise intervention programs involves a comprehensive examination of several factors influencing their long-term viability and impact, as established by previous studies [54,55,56,57]. Employing a multi-dimensional approach, we consider dimensions aligned with the Sustainable Development Goals (SDGs) [58]. The key dimensions included in this assessment comprise: (S1) reporting the percentage of participant adherence and exercise session frequency (aligned with SDGs 3 and 4); (S2) discussing the economic viability of the intervention (in line with SDGs 1 and 8); (S3) describing the potential of the intervention in promoting health equity and inclusivity (corresponding to SDGs 3, 5, and 10); (S4) fostering behavioral and health modifications (linked to SDGs 3 and 4); and (S5) describing the potential of the interventions to generate a positive health and social impact during implementation, associated with SDGs 16 and 17 [59].

## 3. Results

### 3.1. Studies Selection

In the identification stage, our initial search identified 995 potentially eligible papers. Through meticulous screening, we excluded replication articles and studies that did not align with our intervention criteria based on title and abstract analysis. This refined the selection to 48 full-text articles, which underwent rigorous analysis in the screening stage. Applying our core eligibility criteria, 35 papers advanced to the eligibility phase. From this stage, a careful selection process led to the inclusion of 12 SRs papers in our CRs synthesis, as illustrated in Figure 1.

### 3.2. Quality of Evidence

In the methodical assessment of evidence quality utilizing the AMSTAR-2 checklist, the included studies exhibited variable adherence to methodological benchmarks. The assessment encompassed an analysis of 12 systematically reviewed studies [31,60,61,62,63,64,65,66,67,68,69,70]. Among these, four studies were identified as exemplifying high-quality standards [62,64,65,69], while eight studies were categorized as exhibiting critically low quality [31,60,61,63,66,67,68,70]. Only four reviews demonstrated high-quality standards, exemplified by rigorous reporting, transparent review protocols, and comprehensive search strategies [62,64,65,69]. These reviews employed metanalysis techniques, enhancing the synthesis of evidence and contributing to robust conclusions.

All SRs studies assessed the methodological rigor of the included research, applying several types of tools such as Jadad [31,60], PRISMA [61], Cochrane [63,64,68,70], PEDro [62,69], NICE [67], MINOR [66], and Polit and Beck Nursing Studies [65]. Furthermore, a subset of SRs revealed critical shortcomings, particularly in areas such as transparent reporting, complete review protocol adherence, and comprehensive search strategies [31,66,67,68,70]. Among the analyzed studies, the AMSTAR criteria that appeared to be most susceptible to shortcomings were I.2 (characterization and adherence to the complete review protocol), I.4 (comprehensive search protocol), I.7 (list of excluded studies), I.11 (application of a quantitative approach), and I.12 (meta-analysis methodology) (Table 3).

### 3.3. General Characteristics of Included SRs

Table 4 outlines the general characteristics of the selected systematic review (SR) studies, aligning with the PICOTS framework. The publication years of these SRs spanned from 2010 to 2022. Notably, the SRs (without metanalysis) focusing on Chair-Based Exercise (CBE) programs examined only six studies with a participant range of 20 to 82, aged 70 to 90, residing in long-term care facilities [60]. In contrast, the most recent SRs, with the highest participant count (n = 8022), explored exercise effects on physical function, quality of life, and cognition across 23 studies, utilizing meta-analyses for assessment [64].

Concerning the criteria utilized for sample selection and frailty assessment protocols, a variety of criteria were employed in the selected studies. However, it is explicitly mentioned that articles were chosen based on scientifically valid criteria, with Fried’s physical frailty protocol being the predominant choice across the diverse articles [71].

In the selected SR articles, specific exercise types were adopted as inclusion criteria for some studies. These reviews provided detailed descriptions of interventions, including CBE [60], Home-Based Exercise [63], Multicomponent Exercise [31,61], Muscle strength [68], Resistance Training with Elastic Bands [65], and several exercise approaches [62,64,66,67,69,70].

In terms of intervention duration, the SR with the shortest duration covered studies spanning 1 to 24 weeks [65], while the SRs with the longest durations included studies with intervals ranging from 4 to 72 weeks [31]. The number of articles included in each SR varied from six [63] to 47 articles [31]. In general, the SRs presented findings in both qualitative (without metanalysis) and quantitative formats (with metanalysis), offering a comprehensive exploration of diverse exercise interventions.

### 3.4. Excluded SR Studies

In the eligibility phase, 23 out of the initial 35 SRs were excluded based on predefined criteria for this comprehensive review. Some SR has expanded its focus to include therapeutic interventions beyond exercise, such as nutritional and pharmacological approaches [72,73]. Other articles met most criteria but lacked a comprehensive assessment of certain dimensions related to the concept of brain health [74,75,76,77,78,79,80,81]. Specific reviews investigated the combined impact of various exercise modalities [82,83], while one specifically explored physical activity level indicators [84]. Several reviews summarized data on exercise’s influence on brain health indicators, but some incorporated alternative interventions or did not specifically target frail older populations [77,85]. Additionally, some reviews provided concise information on the impact of exercise on older individuals with neurocognitive disorders without specifying whether the study participants were frail [86,87,88,89,90].

### 3.5. Emerging Brain Health Outcomes

This study aimed to discern the critical indicators assessed in the chosen review articles and, amidst the diverse information presented, evaluate the extent to which physical exercise positively influences these variables. Following this study selection, the identified indicators underwent a categorization process, leading to the following dimensions of analysis: Cognitive status, Psychosocial status and (Health) behavior, Health-related Quality of life, Fear of falling, disability and activities of daily life, and Biochemical markers related to cognition and the neuroendocrine system.

#### 3.5.1. Cognitive Status

In the Antony et al. SRs [60], four papers examined mental health, all reporting no negative changes. Two studies utilizing the Mini-Mental State Examination (MMSE) showed significant cognitive improvements at 6 and 12 weeks. In Rossi et al. SRs [69], several cognitive aspects were assessed. Fifteen other cognitive tests demonstrated significant variability in assessment tools. Four studies reported positive exercise effects on processing speed, executive function, and working memory. However, discrepancies were noted in global cognitive improvement, and the Trail Making Test Form A exhibited no significant changes, while the exercise group showed superior performance in Form B.

In Theou et al. SRs [31], neurological function was evaluated by visual stimulus reaction time, auditory stimulus reaction time, coordination, and peripheral sensation. Positive effects were observed in three out of eight cases, including improved visual stimulus reaction time, coordination, and auditory stimulus reaction time. However, no significant impact on peripheral sensation was noted. Cognitive function, assessed using the MMSE in three studies, showed improvement in only one study due to exercise intervention. In Lopez et al. SRs [68], one study evaluated Timed Up and Go test performance with a dual task, showing a significant improvement after 12 weeks of intervention.

#### 3.5.2. Psychosocial Status and (Health) Behavior

In the Antony et al. SRs [60], depression levels were assessed using the Hospital Anxiety and Depression Scale, with one study reporting reduced depression levels at three- and six-months post-exercise intervention. Another study within Antony et al. utilized the Beck Depression Inventory, showing decreased depression levels in both control and exercise groups with no statistically significant differences between them. Dedeyne et al.’s SR study found inconsistent effects on social behavior and depression after exercise intervention in frail individuals [66].

The Saragih et al. SRs reported pooled values from five studies, indicating a significant difference in depression symptoms between the exercise intervention and control groups at 12 and 24 weeks [65]. Finally, Theou et al. SRs also presented several psychosocial state outcomes, indicating positive effects on reducing tiredness related to mobility issues, behavioral problems, safety scores, and self-efficacy and satisfaction with exercise [31]. However, exercise did not significantly impact emotional status, social involvement, or the health belief model. Similar to the findings in Saragih et al.’s SRs, a review conducted by Racey et al. identified low evidence regarding the effects of exercise on self-perceived fatigue [64].

#### 3.5.3. Health-Related Quality of Life

In the Chou SRs, exercise’s impact on HrQoL was explored, and a positive trend was observed in the scores in the mental health component of HrQoL [62]. The Clegg SRs reported only one trial showing an improvement in HrQoL assessed through the EQ5D questionnaire [63]. Saragih et al. found no significant difference in HrQoL between intervention and control groups at 24 weeks across three studies [65]. In Theou et al. SRs, ten studies using various questionnaires, including SF-36, showed exercise interventions improving quality of life in four instances [31]. In Weening-Dijksterhuis’s et al. SRs, two high-quality studies were identified, suggesting a limited impact on depression but varied effects on perceived health and psychosocial function [70]. On the other hand, Racey et al. SRs classified the final selected studies as having low evidence concerning the effects of exercise on HrQoL outcomes [64].

#### 3.5.4. Fear of Falling, Disability, and Activities of Daily Life

In the Antony et al. SRs, fear of falling (FOF) was assessed, but specific levels of change were not reported [60]. The Dedeyne SRs indicated inconsistent effects on FOF following exercise interventions in frail individuals [66]. The Chou SRs explored the impact of exercise on Activities of Daily Living (ADLs) through three trials, revealing that the exercise group exhibited significantly improved ADL performance compared to the control group [62]. However, the Clegg SRs, examining interventions on ADL across four trials [60], reported improvements in ADL in only one trial, with the remaining three trials not showing significant enhancements [63]. The Saragih SRs involved five studies assessing interventions’ impact on ADL and concluded that there was no significant difference in ADL between the intervention and control groups [65]. On the contrary, Theou et al. SRs demonstrated that exercise interventions significantly improved the ADLs, mobility, and disability, assessing these dimensions using different scales based on the evaluation of approximately 20 studies [31].

#### 3.5.5. Biochemical Markers Related to the Neuroendocrine System

The Caldo-Silva SRs included studies that complain about several biochemical markers related to mental health and frailty. These markers included parameters such as glycemia, insulin levels, total cholesterol, triglycerides, High-Density Lipoprotein, Low-Density Lipoprotein, C-Reactive Protein, Vitamin D3, cytokines interleukins (IL) of IL-6, IL-10, IL-1a, IL-1RAcP, Myostatin, Cortisol, Testosterone, Dehydroepiandrosterone, and the Testosterone/Cortisol ratio [61]. However, given the limited number of studies and the diversity of markers examined, the evidence supporting the benefits may be considered very low. In the SRs conducted by Liberman et al., after a different type of exercise training intervention, there were moderate-to-large improvements in inflammatory markers, including increased Brain-Derived Neurotrophic Factor, Insulin-Like Growth Factor 1, and growth hormone [67].

### 3.6. Sustainability Key Elements

All included studies met criterion S1, yet there was inconsistency in reporting participant adherence and exercise frequency across studies. Some provided detailed information on adherence rates, while others lacked comprehensive reporting. Regarding criterion S2, discussions on the economic viability of implementing these types of interventions were generally sparse in the reviewed studies. An exception was Rossi et al. [69], which introduced the impact of public health costs associated with frail conditions. All studies analyzed participants of both genders, but none presented or discussed meta-analysis by gender or the feasibility of implementing this type of intervention for everyone. Concerning criterion S4, varied approaches to fostering behavioral and health modifications were observed, with some studies emphasizing behavioral change strategies more than others. In relation to criterion S5, the potential for interventions to generate positive health and social impacts was inconsistently addressed. Some studies discussed broader societal implications, but overall, this dimension received less attention.

## 4. Discussion

### 4.1. Quality of Evidence and Methodological Rigor

The comprehensive assessment of the systematically reviewed studies revealed a nuanced landscape regarding the quality of the evidence. Among the 12 studies scrutinized through the AMSTAR-2 checklist, the distribution was diverse, with only four studies exemplifying high-quality standards, while nine studies were categorized as exhibiting critically low quality. The incorporation of metanalysis techniques in four high-quality systematic reviews provided a quantitative dimension to the evidence synthesis.

While this SR represents a valuable alternative to traditional meta-analyses for synthesizing evidence [91], it is essential to acknowledge the inherent limitations associated with the narrative synthesis method employed. Noteworthy concerns have been identified in reviews utilizing “narrative synthesis”, raising questions about the robustness of their findings. Chief among these concerns is the often-insufficient description of the methods employed, leading to ambiguity in understanding the synthesis process. Additionally, there may be unclear links between the included data, the synthesis conducted, and the ultimate conclusions drawn. Considering that the NPT (Normalization Process Theory) focuses on evaluating and guiding implementation within RCTs across a range of health and social care areas [92], using it in SRs may improve the way results are presented.

In the evaluation of the surveyed SRs, despite overall adherence to methodological quality benchmarks, critical focal points for improvement were identified. Deficiencies were specifically noted in fulfilling criteria I.2, I.4, I.7, I.11, and I.12 of the AMSTAR-2 checklist [93]. Criterion I.2 highlighted the need for more detailed and transparent reporting of the complete SRs protocol. Partial compliance was observed for criteria I.4 and I.7, emphasizing the necessity of providing a list of excluded studies. This points to the importance of a meticulous and extensive search strategy to minimize bias. Instances of non-compliance with items I.11 and I.12 underscored the need for a more standardized application of a quantitative approach and/or meta-analysis methodology. Similar challenges have been identified in other CRs focusing on physical exercise interventions [94,95].

The multitude of criteria employed for methodological rigor assessment signifies a robust approach [96]. However, this diversity, while reflecting a commitment to established standards, raises concerns about the specificity of these criteria for evaluating studies involving physical exercise interventions. The broad applicability of these criteria may compromise the precision required for assessing the unique nuances of exercise interventions, potentially impacting the selection and interpretation of studies in the SRs. In this sense, the emergence of reliable tools specific to exercise scientists is a critical development, such as the TESTEX scale, which contributes significantly to advancing evidence synthesis in this domain [97]. Its sensitivity allows for a more nuanced and comprehensive evaluation of exercise training trials, effectively addressing potential limitations associated with the use of diverse criteria.

### 4.2. Emerging Brain Health Outcomes

While the exploration of exercise interventions in the context of cognitive status reveals promising trends, it is crucial to temper our enthusiasm with a critical assessment of the evidence base. The current state of evidence, primarily assessed through the number of studies, suggests a low level of confidence in asserting the unequivocal impact of exercise on cognitive skills. This cautious interpretation aligns with the inherent challenges posed by the heterogeneity in study designs, assessment tools, and outcome measures across SRs.

In contrast to the overall positive narrative in certain SRs, notably Racey et al. with meta-analysis [64], the combined evidence unfolds as a mosaic of findings. A key highlight emerges in the form of a moderate trend indicating improvements in simple neuromotor tasks. However, a nuanced complexity arises when scrutinizing the impact on complex motor tasks and other specific areas of cognition, revealing no significant improvements based on the available evidence.

The examination of psychosocial outcomes, including depression levels, yielded mixed but generally positive results. Antony et al. reported reduced depression levels post-exercise [60], aligning with a metanalysis of Saragih et al.’s findings of a significant difference in depression symptoms between exercise and control groups [65]. Notably, some inconsistencies were observed, emphasizing the need for further exploration of exercise’s impact on social behavior and emotional status. The low evidence regarding self-perceived fatigue, as identified by Racey et al., suggests a gap in understanding the holistic psychosocial effects of exercise on frail individuals. Contrary to frail individuals, the patterns of depression, a well-explored aspect of psychosocial well-being [27], are clearer and more established in non-frail populations.

Exercise interventions reduce FOF and ADLs demonstrated mixed outcomes. A poor number of trials included in studies conduct by Clegg et al. SRs [63], the inconsistent results showed in the Antony et al. and Dedeyne et al. SRs [60,66], and ambiguous results presented in the Theou SRs [31], suggests promising potential for exercise interventions, but it shows very low evidence. In this sense, the varied impact on FOF, and ADLs suggests that tailored exercise interventions, considering individual frailty profiles, may yield more consistent positive outcomes. In addition, the ability of exercise to improve the execution of everyday tasks is a neurocognitive paradigm [98], and evidence of the exercise-effect on these indicators is limited in frail older individuals.

The study underscored the importance of immune/inflammatory dysfunction as a central component of frailty, emphasizing its intricate interplay with neuroendocrine damage [99]. Despite numerous studies demonstrating beneficial effects, the evidence regarding the impact of exercise on these markers remains low. The challenge lies in the diverse array of markers investigated under different systems, hindering the transition toward more robust meta-analyses. The high variability in markers across studies complicates the establishment of a cohesive body of evidence [100], making it difficult to draw firm conclusions and emphasizing the need for standardized approaches in exploring the relationship between exercise and neuroendocrine markers.

### 4.3. Sustainability of Exercise Intervention

The examination of sustainability criteria S1 to S5 across the reviewed studies sheds light on both consistency and disparity in reporting practices and considerations. The general results of the analysis of sustainability criteria highlight the need for standardization in reporting practices, economic considerations, and the broader societal impact of exercise interventions for brain health in older populations.

This reporting discrepancy raises significant concerns, given the central role exercise participation-related variables play in ensuring the sustainability of exercise and physical activity programs, aligning with the SDGs [55]. The importance of this dual impact resonates strongly with the SDGs, particularly those concerning health and well-being [101,102]. Recognizing the potential of exercise in promoting sustainable mental health aligns with the broader framework of achieving a sustainable future by focusing on holistic health outcomes.

In a rapidly aging world, comprehending the intricate interplay between physical health, the brain, and the sustainability of exercise programs is crucial. This understanding not only impacts individual well-being but also influences public health policies [103]. Thus, meticulous detailing of these variables in the next original studies or in SRs is crucial, forming the foundation for SRs to provide comprehensive insights into the effectiveness and long-term viability of exercise interventions for older adults with frailty and pre-frailty.

### 4.4. Strengths and Limitations

A major strength lies in the thorough analysis of diverse systematic reviews, allowing for a comprehensive exploration of cognitive, psychosocial, and biochemical dimensions. The inclusion of an exploratory sustainability analysis adds a contemporary perspective, addressing the long-term viability and impact of exercise interventions. However, the study grapples with challenges stemming from the heterogeneity in review methodologies, potentially affecting the overall reliability. The varied outcome measures across reviews pose difficulties in cohesive interpretation. The limited availability of meta-analyses constrains a quantitative assessment of overall effect sizes. The study’s robustness relies on the quality of primary studies within the reviews, introducing variability in evidence strength. Despite these challenges, the synthesis navigates through a mosaic of findings, highlighting a reasonable trend in improving simple neuromotor tasks, depression, and quality of life and shedding light on the complex relationship between exercise and brain health.

### 4.5. Future Directions and Practical Implications

Addressing methodological heterogeneity among systematic reviews is vital for a cohesive evidence base. Future research should prioritize meta-analyses, adopt consistent outcome measures, and explore the temporal aspects of exercise interventions for sustained brain health effects. The practical implications extend to healthcare practitioners, emphasizing the multifaceted impact of exercise on cognitive, psychosocial, and biochemical domains when designing interventions for older individuals. Integrating sustainable practices aligns with the SDG 3, promoting holistic well-being in aging populations. However, limitations in methodological nuances require caution in translating results into real-world clinical decisions and policy formulations, influencing the application of evidence-based interventions and healthcare investments. Recognizing these limitations is crucial for clinicians and policymakers shaping health-related guidelines.

## 5. Conclusions

This systematic review of reviews highlights nuanced findings on exercise interventions for older individuals. While positive trends emerge for certain cognitive and neuromotor aspects, difficulties arise in psychosocial, complex cognitive tasks, and ADL outcomes. The results reveal moderate and promising evidence regarding the impact of exercise on quality of life and depression among older individuals. The impact on biochemical markers remains inconclusive, emphasizing the need for standardized methodologies. The exploration of the interplay between responses and SDGs in the context of physical activity is relatively promising. Consequently, this investigation into this aspect aims to serve as a catalyst, alerting scholars to the potential of integrating this paradigm into future research endeavors. Practical implications underscore the importance of multifaceted exercise programs aligned with the SDGs. Acknowledging methodological nuances is crucial for clinicians and policymakers to translate findings into impactful interventions for aging populations.

## Figures and Tables

**Figure 1 healthcare-11-03160-f001:**
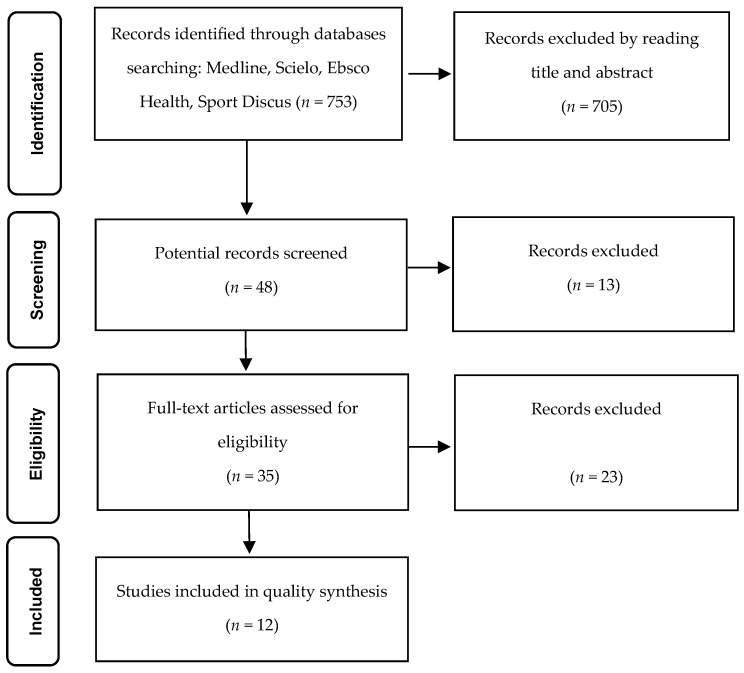
Flow diagram of study selection according to PRISMA guidelines [53].

**Table 1 healthcare-11-03160-t001:** Summary of adopted descriptors for meta-data search.

Physical Exercise
… OR exercise or exercise training OR exercise therapy OR aerobic exercise or aerobic training OR circuit-based exercises OR circuit training OR muscle strength exercises OR muscle strength training or resistance training OR Weight-Lifting exercise program OR muscle stretching exercises OR balance exercises OR flexibility exercises OR multimodal exercise OR multicomponent exercise or combined exercise OR physical activity OR physical fitness programs AND …
Elderly-Frail
… OR frail elders OR frail older adults OR frail older populations OR frailty syndrome OR frailty phenotype OR physical frailty OR frailty outcomes OR frailty status OR frailty state AND …
Systematic review
… OR review OR metanalysis.

**Table 2 healthcare-11-03160-t002:** The main concepts used to guide the meta-search according to the PICOTS statement.

Acronym	Information	Concepts
P	Pre-frail and frail older adults	Adopted an operationalized definition of frailty or standardized criteria to measure frailty previously reported
I	Physical exercise interventions	Planed, repetitive, and structured body movement conducted to improve one or more components of physical fitness (i.e., muscle strength and resistance; flexibility; body composition)
C	Controls or groups of comparison	Without a specific treatment and/or involving isolated or combined physical exercise interventions
O	Brain Health	Indicators on the changing state of the psychosocial, neurocognitive, sensory-motor, social-emotional, sleep, and socio-behavioral domains of an older individual, group of people, or population
T	Time for different types of interventions	Refers to the temporal aspects of exercise interventions, encompassing the duration, frequency, and timing of exercise sessions. It involves specifying how long each exercise session lasts and how often sessions are conducted
S	Community settings or 24 h healthcare and social support centres	Eencompass the environments, contexts, and conditions in which exercise interventions occur. This involves identifying where exercise sessions take place, such as fitness centers, home settings, outdoor locations, or clinical facilities

**Table 3 healthcare-11-03160-t003:** Outcomes of the AMSTAR-2 checklist of quality assessment and their respective items.

	AMSTAR-2 Scale Items																
Author/Year	I.1	I.2	I.3	I.4	I.5	I.6	I.7	I.8	I.9	I.10	I.11	I.12	I.13	I.14	I.15	I.16	GC
Anthony et al., 2013 [60]	Y	N	Y	P	N	Y	P	P	Y	N	N	N	Y	Y	N	N	CL
Caldo-Silva et al., 2021 [61]	Y	Y	Y	P	N	Y	P	P	Y	N	N	N	Y	Y	N	Y	CL
Chou et al., 2012 [62]	Y	N	Y	P	N	Y	P	P	Y	N	Y	Y	Y	Y	N	Y	H
Clegg et al., 2012 [63]	Y	N	Y	P	N	Y	P	P	Y	N	Y	Y	N	N	N	Y	CL
Racey et al., 2021 [64]	Y	Y	Y	P	Y	Y	P	P	Y	N	Y	Y	Y	Y	Y	Y	H
Saragih et al., 2022 [65]	Y	Y	Y	P	Y	Y	P	P	Y	N	Y	Y	Y	Y	Y	Y	H
Dedeyne et al., 2017 [66]	Y	Y	Y	P	N	Y	P	P	Y	N	N	N	Y	N	N	Y	CL
Liberman et al., 2017 [67]	Y	N	Y	N	N	N	P	P	Y	N	N	N	N	N	N	Y	CL
Lopez et al., 2018 [68]	Y	N	Y	P	Y	Y	P	P	Y	N	N	N	N	Y	N	Y	CL
Rossi et al., 2021 [69]	Y	Y	Y	P	Y	Y	P	P	Y	N	Y	Y	Y	Y	Y	Y	H
Theou et al., 2011 [31]	Y	N	Y	P	Y	Y	P	P	Y	N	N	N	N	Y	N	Y	CL
Weening et al., 2011 [70]	Y	N	Y	P	N	N	P	P	Y	N	N	N	N	Y	N	Y	CL

Notes: I.1: Research questions following PICOTS; I.2: Explicit establishment of the SRs protocol; I.3: Inclusion of a thorough and SR search; I.4: Appropriate technique to appraise the scientific quality of studies; I.5: Use of suitable methods for combining study findings; I.6: Evaluation of publication bias; I.7: Scientific quality in formulating conclusions; I.8: Appropriate methods for the statistical combination of results; I.9: Presence of publication bias; I.10: Methods for the statistical combination of results; I.11: Impact of the risk of bias on SR results; I.12: Assess the impact of the risk of bias; I.13: Consideration of the impact of funding sources; I.14: Impact of conflicts of interest, I.15: Satisfactory technique to appraise scientific quality in the SRs conclusion; I.16: Evaluation of the scientific quality of the SRs conclusion.

**Table 4 healthcare-11-03160-t004:** Characteristics of the selected SR studies following the PICOTS statement.

Author (Year) Reference	Total and Range of Study Participants/Settings	Type of Exercise Interventions	Comparison	Brain Health Outcomes	Time of Intervention (Range)
Anthony et al. (2013) [60]	A total of 251 frail older adults, 20 to 82 male and female, ranging in age from 70 to 99 years old, lived in community and health care facilities	Chair-based exercises versus non-exercise (standard care) controls	Supervised group classes in facilities and home-based exercise (2 to 3 times per week, 45 to 60 min sessions)	Cognitive status, depression, FOF	6 to 24 weeks
Caldo-Silva et al. (2021) [61]	A total of 379 frail older adults, 60 to 100 male and female, age ranging from 73 to 86 years old, living in the community	Multi-component exercise programs vs. non-exercising controls	Supervised group class (3 to 5 times per week, 45 to 60 min of time)	Biomarkers related to neurocognition (BDNF, interleukins)	12 to 72 weeks
Chou et al. (2012) [62]	A total of 1068 frail older adults, 10 to 158 male and female, ranging in age from 75.3 to 86.8 years old, lived in the community	Exercise-training, single or multiple, including flexibility, resistance, aerobics, coordination, balance, and Tai-Chi vs. non-exercise controls	Supervision is either exercised in facilities, communities, or at home. 60-to-90-min sessions, repeated daily or weekly	HrQoL, FOF, and ADL performance	3 to 48 weeks
Clegg et al. (2012) [63]	A total of 987 frail older adults, 61 to 486 male and female, ranging in age from 78 to 88 years old, lived in the community	Home-based exercise interventions	Supervised exercise 3 times per week	HrQoL, ADL performance	6 to 64 weeks
Racey et al. (2021) [64]	A total of 8022 frail older adults, 280 to 1635 male and female, aged 69 to 84 years old	Group-based muscle, aerobic, and clinical exercise interventions	Supervised exercise 1 to 4 times per week	ADL, cognitive function, fatigue, HrQoL, and FOF	6 to 96 weeks
Saragih et al. (2022) [65]	A total of 1294 frail older adults, 9 to 126 male and female, ranging in age from 65 to 86 years old, lived in the community	Muscle-strength exercises vs. controls: non-exercising	Supervision exercise program in facilities exercised 45 to 90 min/sessions, 2 to 3 times per week	Depression, FOF, and ADL performance, HrQoL	12 to 98 weeks
Dedeyne et al. (2017) [66]	A total of 1202 frail older adults, 31 to 326 male and female, ranging in age from 71-to-83 years old, lived in the community	Multidomain modalities: chair yoga, resistance training, strength exercises	Supervise a group-class exercise program, varying in duration from 45 to 90 min/sessions, 2 to 3 times per week x non-exercising controls	Psychosocial and (Health) behavior	12 to 36 weeks
Liberman et al. (2017) [67]	A total of 282 frail older adults, 15 to 52 male and female, ranging in age from 65-to-72 years old, lived in the community	Muscle-strength, aerobic, combined, and body vibration exercises.	Supervise a group-class exercise program, varying in duration from 30 to 90 min/sessions, single sessions to 3 times per week x non-exercising controls	Pro- and anti-inflammatory cytokines, neurotrophic factors	6 to 32 weeks
Lopez et al. (2018) [68]	A total of 1812 frail older adults, 23 to 616 male and female, age ranging from 65-to-? (not refer) years old, living in community	Resistance and muscle-strength training (RMST)	Classic RMST protocols comprise a number of sets (1–3), repetitions (6–15), and 45–90 min by session	Cognition status of dual tasks	10 to 48 weeks
Rossi et al. (2021) [69]	A total of 325 frail older adults; 22 to 100 male and female, age ranging from 65-to-84 years old, living in the community	RMST, multicomponent, high-speed resistance exercise vs. non-exercising controls	Supervision of a group class exercise program, varying in duration from 40 to 90 min/sessions, single sessions 1 to 3 times per week	Cognitive functions	12 to 24 weeks
Theou et al. (2011) [31]	A total of 325 frail older adults; 13 to 188 male and female, age ranging from 65-to-84 years old, living in community and home cares	Multicomponent training program	Supervised group class exercise, a frequency of 2–3 times per week, 30 to 90 min of time session	Neurological, psychosocial, HrQoL, and ADL performance	4 to 72 weeks
Weening et al. (2011) [70]	A total of 325 frail older adults, 20 to 981 male and female, age ranging from 65-to-84 years old, living in community	Balance, aerobics, flexibility, and strength, combined with functional training exercises	Supervised group class exercise, a frequency of 2–3 times per week, 45 to 60 min of time session	HrQoL	4 to 72 weeks

Notes: HrQoL = Health Related Quality of Life; ADL = Activities of daily life; FOF = Fear of falling; RMST = Resistance and Muscle-strength training.

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
