# Peer review of "Impact of Exercise Interventions on Sustained Brain Health Outcomes in Frail Older Individuals: A Comprehensive Review of Systematic Reviews"

_healthcare, 2023, doi:10.3390/healthcare11243160_

Round 1
Reviewer 1 Report
Comments and Suggestions for Authors
This investigation conducts a comprehensive review of systematic reviews, selecting studies in alignment with specific inclusion and exclusion criteria. While conventional systematic review methodologies typically exclude secondary reviews based on exclusion criteria, this approach is justifiable within the context of a comprehensive review. Nevertheless, there is a necessity for an explicit justification within the introductory section as to why a comprehensive review of systematic reviews was undertaken, to reconcile with the general expectations surrounding the synthesis of systematic reviews post exclusion.
Methods:
The methodological approach appears to conform with the established systematic review procedures, suggesting meticulous adherence to scholarly research standards.
Results:
The section '3.4. Excluded SR Studies' addresses exclusions based on criteria distinct from those discussed in '2.3. Exclusion Criteria,' indicating a potential misalignment with the initially set standards, which are not thoroughly explained. Furthermore, there is a notable discrepancy between Figure 1, which suggests that 23 out of 35 studies were excluded for not meeting eligibility, and the narrative content, which asserts that 24 out of 37 studies were excluded.
Discussion:
Despite the research being well-executed and the results robust, the discussion section is somewhat lacking in substance. Considering that the study is a comprehensive review of systematic reviews, the discussion should offer a more thorough and nuanced examination of the gathered data and its implications for exercise interventions aimed at improving cognitive health in the frail elderly demographic.
Author Response
Reviewer #01 - Comments and Suggestions for Authors
REV#01) This investigation conducts a comprehensive review of systematic reviews, selecting studies in alignment with specific inclusion and exclusion criteria. While conventional systematic review methodologies typically exclude secondary reviews based on exclusion criteria, this approach is justifiable within the context of a comprehensive review. Nevertheless, there is a necessity for an explicit justification within the introductory section as to why a comprehensive review of systematic reviews was undertaken, to reconcile with the general expectations surrounding the synthesis of systematic reviews post exclusion.
Authors: Thank you for your suggestion. The introduction section was changed taking into account your recommendations. To strengthen the focus on the "concrete and more direct objective", we added a study hypothesis, changed the results and conclusions in the summary and increased the discussion and conclusion throughout the text.
REV#01) Methods: The methodological approach appears to conform with the established systematic review procedures, suggesting meticulous adherence to scholarly research standards.
Authors: Thank you for your observation.
REV#01) Results: The section '3.4. Excluded SR Studies' addresses exclusions based on criteria distinct from those discussed in '2.3. Exclusion Criteria,' indicating a potential misalignment with the initially set standards, which are not thoroughly explained. Furthermore, there is a notable discrepancy between Figure 1, which suggests that 23 out of 35 studies were excluded for not meeting eligibility, and the narrative content, which asserts that 24 out of 37 studies were excluded.
Thank you for your consideration. The introduction section 2.4 was adjusted taking into account your recommendations.
REV#01) Discussion: Despite the research being well-executed and the results robust, the discussion section is somewhat lacking in substance. Considering that the study is a comprehensive review of systematic reviews, the discussion should offer a more thorough and nuanced examination of the gathered data and its implications for exercise interventions aimed at improving cognitive health in the frail elderly demographic.
Thank you for your consideration. The discussion section 2.4 was adjusted taking into account your recommendations. In fact, it was necessary to make it very clear which indicators or dimensions make up brain health that actually have concrete evidence of changes over time.
Reviewer 2 Report
Comments and Suggestions for Authors
This comprehensive review of systematic reviews aims to analyse the influence of exercise on the sustainability of cognitive health outcomes in frail and pre-frail older adults. The topic is socially interesting and, in addition, the authors have used a large number of literature references (more than 100), around 70% from the last 10 years.However, there are weaknesses in this work:
- The number of reviews analysed should be clarified and be consistent throughout the text. In the abstract, 11 studies are mentioned (line 42), but 4 of high quality and 9 of low quality (lines 42 and 43). In addition, the flow chart lists 12 (Figure 1) and 14 references (line 222, studies 33-36,55-64), Table 3 lists 11 studies analysed with AMSTAR-2 and the Conclusions list 11 studies (line 548). References 36 and 61 are the same, so I think there are 13 references, not 11 or 14.
- In the Introduction section, references from Results (lines 98-100) are used. It is not appropriate to support a topic in the Introduction with part of the references that will appear in the Results.
- The first time an abbreviation appears, it is necessary to point out what it means, even if it is known (line 114, SDGs).
- In the Results section (section 3.1), the data reported do not coincide with those appearing in the flow chart (Figure 1).
- Section 3.2. discusses the methodological quality analysis instruments (lines 225-230). Study number 33 is not reflected.
- Table 3 lacks the analysis of studies 57 and 63. In addition, the way the first author is referenced does not correspond to the Bibliography section. Finally, the tables should be self-explanatory, and the meaning of each item of the AMSTAR-2 is missing in the legend.
- Nor does study 33 appear in the description of the type of exercise intervention (lines 263-271).
- I do not understand the importance of section 3.4. Following the PRISMA guidelines, the excluded studies and their reasons should appear in the flow chart. In addition, the data for excluded studies in Figure 1 do not match the data in this section (line 281).
- The way of referencing the different studies in the Results section is not appropriate. For example, Antony SR study (line 305), or Antony study (line 327) or Antony SR paper (line 329), should be Antony et al., or Antony et al. SR study, or Antony et al. Study. There are quite a few references to specific studies that should be corrected.
- FOF (line 378) has already been defined previously (line 336).
- I think the abbreviations in section 3.5.5. are not necessary (they are not used again).
- I don’t understand what the authors want to tell us with section 4.1. that is different from what appears in Results.
- The Discussion seems to me to add nothing new to what has already been published. How does what the authors say differ from what is expressed in the references used between lines 507-519? It seems to me that nothing is specified about exercise programmes in relation to the United Nation's SDGs, for example, which would be one of the strengths of the work according to the authors.
- The conclusions are generic and undefined.
Comments on the Quality of English LanguageMinor editing of English language required
Author Response
Reviewer #02 Comments
REV#02) This comprehensive review of systematic reviews aims to analyse the influence of exercise on the sustainability of cognitive health outcomes in frail and pre-frail older adults. The topic is socially interesting and, in addition, the authors have used a large number of literature references (more than 100), around 70% from the last 10 years. However, there are weaknesses in this work:
REV#02) The number of reviews analysed should be clarified and be consistent throughout the text. In the abstract, 11 studies are mentioned (line 42), but 4 of high quality and 9 of low quality (lines 42 and 43). In addition, the flow chart lists 12 (Figure 1) and 14 references (line 222, studies 33-36,55-64), Table 3 lists 11 studies analysed with AMSTAR-2 and the Conclusions list 11 studies (line 548). References 36 and 61 are the same, so I think there are 13 references, not 11 or 14.
Authors) Dear reviewer, thank you for your utmost attention to this point. In fact, after installing a new version of Mendeley on my PC, there was some noise here that caused this confusion in the references. Regarding the number of studies included, excluded and those that remained in the final review, there was also an adjustment, taking into account that there was one study (Racey study) that was omitted.
REV#02) - In the Introduction section, references from Results (lines 98-100) are used. It is not appropriate to support a topic in the Introduction with part of the references that will appear in the Results.
Authors) Thank you for your recommendation. Inappropriate references have been removed
- REV#02). The first time an abbreviation appears, it is necessary to point out what it means, even if it is known (line 114, SDGs).
Authors) Thank you for your recommendation. These and other abbreviations have been duly corrected.
REV#02) - In the Results section (section 3.1), the data reported do not coincide with those appearing in the flow chart (Figure 1).
Authors) Thank you for your observation. This section was completely revised, including the cited references were checked after correcting the error in the reference editor.
REV#02) - Section 3.2. discusses the methodological quality analysis instruments (lines 225-230). Study number 33 is not reflected.
Authors) Thank you for your observation. This section was completely revised, including the cited references were checked after correcting the error in the reference editor.
REV#02) - Table 3 lacks the analysis of studies 57 and 63. In addition, the way the first author is referenced does not correspond to the Bibliography section. Finally, the tables should be self-explanatory, and the meaning of each item of the AMSTAR-2 is missing in the legend.
Authors) Thank you for your observation. This section was completely revised, including the cited references were checked after correcting the error in the reference editor. Also, the and the meaning of each item of the AMSTAR-2 was introduced in the legend of table 3
REV#02) - Nor does study 33 appear in the description of the type of exercise intervention (lines 263-271).
Authors) Thank you for your observation. This section was completely revised, including the cited references were checked after correcting the error in the reference editor.
REV#02) - I do not understand the importance of section 3.4. Following the PRISMA guidelines, the excluded studies and their reasons should appear in the flow chart. In addition, the data for excluded studies in Figure 1 do not match the data in this section (line 281).
Authors) I understand your position, However, the methodology for this article suggests that we report the excluded studies. In this sense I will keep it, if you don't mind. The option of not reporting in the table is due to a lack of physical space.
REV#02) - The way of referencing the different studies in the Results section is not appropriate. For example, Antony SR study (line 305), or Antony study (line 327) or Antony SR paper (line 329), should be Antony et al., or Antony et al. SR study, or Antony et al. Study. There are quite a few references to specific studies that should be corrected.
Authors) Thanks for observation. The language for presentation and discussion of all articles was standardized.
REV#02) - FOF (line 378) has already been defined previously (line 336).
Authors) Thanks for observation. I make some changes in the acronyms
REV#02) - I think the abbreviations in section 3.5.5. are not necessary (they are not used again).
Authors) Thanks for observation. In fact, the markers are mentioned only in this paragraph.
REV#02) - I don’t understand what the authors want to tell us with section 4.1. that is different from what appears in Results.
REV#02) - The Discussion seems to me to add nothing new to what has already been published. How does what the authors say differ from what is expressed in the references used between lines 507-519? It seems to me that nothing is specified about exercise programmes in relation to the United Nation's SDGs, for example, which would be one of the strengths of the work according to the authors.
Authors) You're right, the content of the old discussion was redundant and didn't bring anything new. However, the discussion was redone, bringing some critical points that are more interesting for the reader.
REV#02) The conclusions are generic and undefined.
Authors) Thanks for observation. Like the discussion section, the conclusion section brought some of the most apparent or evident results found in the articles, which can, to a certain extent, bring new reflections and directions to readers.
Reviewer 3 Report
Comments and Suggestions for Authors
The abstract clearly states the purpose of the review and the methods used, which includes a comprehensive review (CR) of systematic reviews approach. However, it should specify the databases searched, give a more concise and directly state the primary objectives of the review and the conclusions could be strengthened by suggesting specific areas for future research or potential interventions based on the findings
The introduction effectively sets the stage for the importance of studying the interplay between physical well-being and cognitive vitality in aging populations. Although, it should be more concise, particularly in summarizing the various frailty assessment tools and their relevance to the study. It would benefit from a clearer and more direct statement of the study’s primary research question or hypothesis and should explicitly state how the current study will add to the existing body of knowledge
The methodology section provides a clear and systematic approach to conducting a comprehensive revie. The exclusion criteria are well-defined, but the rationale behind each criterion could be further explained to understand the scope and limitations of the review. In the quality assessment section, there is a mention of excluding articles with critical quality evidence if "essential details were absent," which could be more specific about what constitutes "essential details."
No problems detect in results section
The discussion comprehensively addresses the quality of evidence and highlights the strengths of the studies included in the review. It could benefit from a more detailed exploration of how the identified methodological limitations might have influenced the review's overall conclusions and more insight into the implications of the study findings for clinical practice and policy-making.
The conclusion could be strengthened by explicitly stating the main findings before discussing their implications and it needs for a more direct commentary on the practical implications of these findings for health practitioners and policymakers.
Author Response
REV#3) The abstract clearly states the purpose of the review and the methods used, which includes a comprehensive review (CR) of systematic reviews approach. However, it should specify the databases searched, give a more concise and directly state the primary objectives of the review and the conclusions could be strengthened by suggesting specific areas for future research or potential interventions based on the findings.
Authors) Thank you for your considerations: The abstract section was changed taking into account your recommendations.
REV#3) The introduction effectively sets the stage for the importance of studying the interplay between physical well-being and cognitive vitality in aging populations. Although, it should be more concise, particularly in summarizing the various frailty assessment tools and their relevance to the study. It would benefit from a clearer and more direct statement of the study’s primary research question or hypothesis and should explicitly state how the current study will add to the existing body of knowledge
Authors: Thank you for your considerations. The introduction section was changed taking into account your recommendations. To reinforce the focus on the "concrete and more direct objective", we added a study hypothesis, changed the results and conclusions in the summary and increased the discussion and conclusion throughout the text.
REV#3) The methodology section provides a clear and systematic approach to conducting a comprehensive revie. The exclusion criteria are well-defined, but the rationale behind each criterion could be further explained to understand the scope and limitations of the review. In the quality assessment section, there is a mention of excluding articles with critical quality evidence if "essential details were absent," which could be more specific about what constitutes "essential details."
Authors) the essential details that caused the reviews to be excluded were highlighted
REV#3) No problems detect in results section. The discussion comprehensively addresses the quality of evidence and highlights the strengths of the studies included in the review. It could benefit from a more detailed exploration of how the identified methodological limitations might have influenced the review's overall conclusions and more insight into the implications of the study findings for clinical practice and policy-making.
Authors) Thank you for observation. This detail was added in the new version of the discussion section.
REV#3) The conclusion could be strengthened by explicitly stating the main findings before discussing their implications and it needs for a more direct commentary on the practical implications of these findings for health practitioners and policymakers.
Authors) Thank you for your considerations. The conclusion section was changed taking into account your recommendations;
Round 2
Reviewer 2 Report
Comments and Suggestions for Authors
I believe that the authors have observed the suggestions and recommendations given by the reviewers, which has benefited the quality of the manuscript.
Overall, the manuscript is clear and well-structured, being an interesting comprehensive review of systematic reviews on the influence of exercise on the sustainability of cognitive health outcomes in frail and pre-frail older adults. The conclusions are consistent with the findings.